# xDDPM: EXPLAINABLE DENOISING DIFFUSION PROBABILISTIC MODEL FOR SCIENTIFIC MODELING

**Qianru Zhang**
Department of Computer Science
The University of Hong Kong
{qrzhang}@cs.hku.hk

**Chenglei Yu**
Department of Computer Science
Westlake University
{yuchenglei}@westlake.edu.cn

**Yudong Yan**
Tsinghua University
{yudongyan167}@gmail.com

**Xiangyu Kuang**
University of California, Irvine
{xkuang2}@uci.edu

**Yi MA**
The University of Hong Kong
{yima}@eecs.berkeley.edu

**Yuansheng cao**
Tsinghua University
{yscao}@tsinghua.edu.cn

**Siu-Ming Yiu**[*]
The University of Hong Kong
{smyiu}@cs.hku.hk

**Tailin Wu**[†]
Westlake University
{tailin}@cs.stanford.edu

## ABSTRACT

In recent years, diffusion models have emerged as powerful tools for generatively predicting high-dimensional observations across various scientific and engineering domains, including fluid dynamics, weather forecasting, and physics. Typically, researchers not only want the models to have faithful generation, but also want to *explain* these high-dimensional generations with accompanying signals such as measurements of force, currents, or pressure. However, such explainable generation capability is still lacking in existing diffusion models. Here we introduce Explainable Denoising Diffusion Probabilistic Model (xDDPM), a simple variant to the standard DDPM that enables the generation of samples in an explainable manner, focusing solely on generating components that are pertinent to the given signal. The key feature of xDDPM is that it trains the denoising network to exclusively denoise these relevant parts while leaving non-relevant portions noisy. It achieves this by incorporating an Information Bottleneck loss in its learning objective, which facilitates the discovery of relevant components within the samples. Our experimental results, conducted on two cell dynamics datasets and one fluid dynamics dataset, consistently demonstrate xDDPM's capability for explainable generation. For instance, when provided with force measurements on a jellyfish-like robot, xDDPM accurately generates the relevant pressure fields surrounding the robot while effectively disregarding distant fields.

## 1 INTRODUCTION

Generative models like diffusion models (Ho et al., 2020) are valuable for complex tasks in science and engineering, learning from high-dimensional distributions to predict dynamics in fluid dynamics (Cachay et al., 2023), weather forecasting (Price et al., 2023), molecular dynamics (Wu et al., 2022), and more. However, interpreting the increasingly complex generated output (often $10^3$ to $10^6$ dimensions) is challenging. Researchers seek to understand the relationship between generated components and accompanying low-dimensional signals, such as pressure measurements in fluid

---

[*]Corresponding Author.
[†]Corresponding Author.

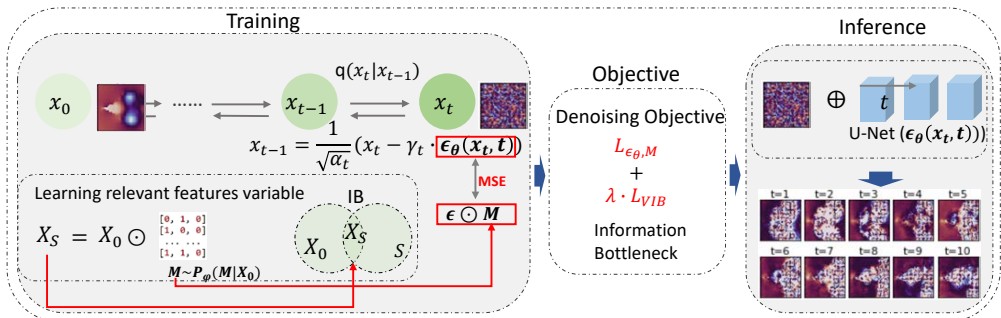

Figure 1: Architecture of our proposed **xDDPM** framework. Our method consists of three main components. The first component involves the training process using the Information Bottleneck (IB) mechanism, which is constrained by the objective function (described in the second component). The third component illustrates the inference process.

dynamics or brain signals in behaviors. Building a diffusion generative model that faithfully learns and explains high-dimensional distributions is crucial.

Explainable generation tasks pose significant challenges in linking relevant components of high-dimensional generated samples to low-dimensional signals. Existing attribution methods, such as Schulz et al. (2020); Selvaraju et al. (2017), are designed for classification tasks with prepared datasets, making them unsuitable for dynamic generation tasks without available low-dimensional signals at inference time. Additionally, the high dimensionality (up to $10^3$ to $10^6$) of the samples further complicates the identification of relevant components. To the best of our knowledge, no diffusion model research has addressed this crucial question.

We present xDDPM, an innovative approach to address the task of explainable generation. Unlike previous methods, xDDPM trains the denoising network to selectively denoise the signal-relevant parts of the sample, allowing the non-relevant parts to remain noisy. This explicit association between generation and the given signal is achieved by incorporating an Information Bottleneck (IB) module into the existing DDPM method (Ho et al., 2020). The IB module learns to identify the relevant components for the signal, which are then used to modify the noise target during training. Consequently, xDDPM's denoising network can generate samples containing only the relevant signal parts while keeping the non-relevant parts noisy during inference.

We make the following contributions: 1) Introducing xDDPM, a diffusion-based model for explainable generation. It incorporates an Information Bottleneck to modify the denoising target, generating samples with only signal-relevant components. 2) Providing valuable datasets, including cell dynamics and fluid dynamics, with over 45,000 videos and 1 million images. These datasets facilitate research in explainable generation. 3) Conducting extensive experiments, outperforming seven state-of-the-art baselines on the datasets. For implementation and results, visit https://anonymous.4open.science/r/xDDPM. xDDPM enables interpretable scientific modeling and sets a new performance standard.

## 2 RELATED WORK

**Diffusion Probabilistic Models (DM)**. Ho et al. (2020); Kingma et al. (2021) are leading techniques for density estimation and high-quality sample generation. They leverage image-like data using a UNet neural backbone Ronneberger et al. (2015); Ho et al. (2020); Dhariwal & Nichol (2021). A reweighted objective in training improves synthesis quality. Another approach for image generation is GAN-based methods, such as infoGAN Creswell et al. (2018); Chen et al. (2016), which generates realistic samples and maximizes mutual information between latent variables and output. However, these methods lack explanations for generated samples. **Denosing Diffusion Probabilistic Model (DDPM) for AI for Science** and **Explanation for Artificial Intelligence** are put into Appendix.

Table 1: A comprehensive performance comparison of all methods on datasets of Wetting, Tension, and Fluid. For cell Wetting and Tension datasets, we independently provide values of area or circumference as signals, and the goal is to learn to generate the dynamics video where only the pixels relevant to the signal are shown, and irrelevant pixels remain noisy. For the Fluid dataset, the method needs to generate the fluid trajectory while identifying which components of the pressure field are relevant to the force measurement on the boundary. We see that our xDDPM consistently outperforms the baselines by a wide margin across all datasets.

| | Wetting | | | | Tension | | | | Fluid | |
| | Area | | Circumference | | Area | | Circumference | | Force | |
| Method | IoU | Sensitivity-n | IoU | Sensitivity-n | IoU | Sensitivity-n | IoU | Sensitivity-n | Correlation | Sensitivity-n |
|---|---|---|---|---|---|---|---|---|---|---|
| Random | 0.2389 | 0.0315 | 0.0717 | 0.1338 | 0.3297 | 0.0094 | 0.1744 | 0.0467 | 0.0003 | 0.2432 |
| Gradient | 0.2147 | 0.0181 | 0.0882 | 0.0153 | 0.5309 | 0.0015 | 0.2636 | 0.0044 | 0.0064 | 0.0051 |
| Saliency | 0.2811 | 0.0290 | 0.0892 | 0.1252 | 0.4896 | 0.0151 | 0.2414 | 0.0308 | 0.0327 | 0.1806 |
| GuidedBP | 0.1033 | 0.0734 | 0.0238 | 0.1065 | 0.0350 | 0.0368 | 0.1399 | 0.0111 | 0.0033 | 0.2021 |
| GuidedCAM | 0.0601 | 0.0199 | 0.0221 | 0.0980 | 0.0399 | 0.0081 | 0.0734 | 0.0085 | 0.0085 | 0.1974 |
| SmoothGrad | 0.1755 | 0.0049 | 0.0612 | 0.1677 | 0.4691 | 0.0211 | 0.3503 | 0.0451 | 0.0093 | 0.0229 |
| GuidedGrad | 0.0723 | 0.0290 | 0.0350 | 0.1017 | 0.1272 | 0.0114 | 0.0514 | 0.0392 | 0.0285 | 0.2182 |
| **xDDPM (ours)** | **0.4590** | **0.3114** | **0.1197** | **0.1713** | **0.6689** | **0.1638** | **0.3699** | **0.1312** | **0.3343** | **0.3881** |

## 3 METHOD

In this section, we detail our method of Explainable Denoising Diffusion Probabilistic Model (xDDPM). We first introduce the problem setup in Section 3.1. In Section 3.3, we detail our xDDPM method, including its learning objective, model architecture, and inference method.

### 3.1 PROBLEM DEFINITION

Consider that we have high-dimensional data samples[1] $\{x^{(n)}\}_{n=1}^N$ of the variable $X \in R^D$. Accompanying each $x^{(n)}$ is the low-dimensional signal $s^{(n)} \in R^d$ for signal variable $S$. Here, instead of the standard task of learning the distribution[2] $P_X(x)$ for generating $X$, we are interested in learning a distribution $P_W(w)$ for generating the *explainable* variable $W$, such that $W$ is as faithful to $X$ as possible while containing exclusively *relevant* features to $S$. Concretely, we want to learn a distribution $P_W(w)$ for sampling $W$, where $W = g(X, S) \in R^D$ has the same dimension as $X$, such that the relevant features of $W$ w.r.t. $S$ obey the same distribution as $X$ and the non-$S$-relevant features obey a Gaussian distribution. Note that during inference time, we do not have the accompanying signal $S$ for generating $W$. Thus, it is impossible to first generate $X$ and then use attribution methods to find which features of $X$ are relevant to $S$ and then produce $W$. Therefore, the model for generating $W$ must be learned at training time.

### 3.2 EXPLAINABLE DENOISING DIFFUSION PROBABLISTIC MODELS (XDDPM)

**The DDPM method.** To tackle the above task, we build upon the recent advances of Denoising Diffusion Probablistic Models (DDPMs) Ho et al. (2020). DDPM is an elegant method to learn high-dimensional distributions $P_X(x)$ given data samples $\{x^{(n)}\}_{n=1}^N$. Concretely, DDPM consists of a forward process that adds $t$ steps of Gaussian noise to the data sample[3] $x_0$ to obtain a noisy sample $x_t$: $x_t = \sqrt{\bar{\alpha}_t}x_0 + \sqrt{1 - \bar{\alpha}_t}\epsilon$, where $\epsilon \sim \mathcal{N}(0, I)$ has the same dimension as $x_0$ and $\bar{\alpha}_t$ is a predefined schedule. Since the summation of Gaussian is also a Gaussian, the $t$ steps of adding Gaussian noise is equivalent to adding a single Gaussian $\sqrt{1 - \bar{\alpha}_t}\epsilon$. After $T$ steps of forward process, $x_T$ approximates a standard Gaussian distribution. In the reverse process, the DDPM learns a denoising model $\epsilon_\theta(x_t, t)$ that aims to revert the forward process:

$$x_{t-1} = \frac{1}{\sqrt{\alpha_t}}\left(x_t - \frac{\sqrt{1 - \alpha_t}}{\sqrt{1 - \bar{\alpha}_t}}\epsilon_\theta(x_t, t)\right) + \sigma_t\eta, t = T, ...1.$$

Here $\alpha_t, \sigma_t$ are pre-defined schedule and $\eta \sim \mathcal{N}(0, I)$. To learn $\epsilon_\theta(x_t, t)$, DDPM use the denoising objective

$$L_{\epsilon_\theta} = ||\epsilon - \epsilon_\theta(\sqrt{\bar{\alpha}_t}x_0 + \sqrt{1 - \bar{\alpha}_t}\epsilon, t)||_2^2, \epsilon \sim \mathcal{N}(0, I) \tag{1}$$

---

[1]Here we use capital letters (e.g., $X$) to denote random variables and lowercase letters (e.g., $x$) to denote their instances. The lowercase letters with superscript (e.g., $x^{(n)}$) denote data samples.

[2]Sometimes, we may be interested in learning a condition distribution $P_{X|C}(x|c)$ for optional condition variable $C$. For notation simplicity, we ignore such conditions and only add it as needed.

[3]Here the subscript $t$ in $x_t$ denotes the denoising step.

which aims to predict the added noise $\epsilon$ based on the noisy sample $x_t = \sqrt{\bar{\alpha}_t}x_0 + \sqrt{1 - \bar{\alpha}_t}\epsilon$. For more details about DDPM, see Appendix A.2.

## 3.3 EXPLAINABLE DENOISING DIFFUSION PROBABLISTIC MODELS (XDDPM)

**The xDDPM method.** Although the above DDPM can learn complicated distribution $P_X(x)$ given data samples, it is insufficient to learn the distribution $P_W(w)$ for the explainable variable $W$ since there are no ground-truth data $W$ to learn the distribution from. Our key insight is that the relevance discovery of $X$ to the signal $S$ is essentially finding the *minimal sufficient* information contained within $X$ for predicting $S$, and the Information Bottleneck Tishby et al. (2000) provides the exact principle and technique we need for extracting such minimal sufficient information. Specifically, we consider a noisy representation $X_S$ of $X$ where $X_S$ only contains the relevant features of $X$ w.r.t. $S$. To learn such a representation, we employ the following Information Bottleneck (IB) objective:

$$L_{\text{IB}} = I(X; X_S) - \beta \cdot I(X_S; S) \tag{2}$$

Here $I(\cdot; \cdot)$ denotes mutual information, and $\beta$ is a hyperparameter. By minimizing the above IB objective, it encourages $X_S$ to contain as much information as possible for predicting $S$, while retaining as much little information as possible about $X$, thus encouraging $X_S$ to contain the minimal sufficient information of $X$ for predicting $S$.

We define $X_S$ as $X$ multiplied with a continuous mask $M \in [0, 1]^D$, where the mask values indicate the per-feature relevance: $X_S = X \odot M$ Here $\odot$ denotes element-wise multiplication. To obtain $M$, we use an encoder network $p_\varphi(M|X)$ with learnable parameters $\varphi$. Since the IB objective is intractable, we employ the deep Variational Information Bottleneck (VIB) Alemi et al. (2016) to minimize its upper bound:

$$L_{\text{VIB}} = \mathbb{E}_{X_S \sim p_\varphi(X_S|X)} \left[ \log \frac{p_\varphi(X_S|X)}{q_\varphi(X_S)} - \beta \log q_\varphi(S|X_S) \right] \tag{3}$$

The first term provides an upper bound for $I(X; X_S)$ and the second term provides an upper bound for $-\beta \cdot I(X_S; S)$ in Eq. 2. Here the encoder $p_\varphi(X_S|X)$, the prior distribution $q_\varphi(X_S)$, and the decoder $q_\varphi(S|X_S)$ are all parameterized by learnable neural networks. Minimizing the above $L_{\text{VIB}}$ objective encourages finding an encoder $p_\varphi(M|X)$ for the mask $M$ so that the $X_S = X \odot M$ approximately extracts the minimal necessary information of $X$ for $S$.

Given the above IB module, how can we encourage the DDPM to learn to generate the explainable variable $W$ relevant to $S$? Note that the objective $L_{\epsilon_\theta}$ in Eq. 10 encourages the denoising network $\epsilon_\theta$ to denoise all features of $x_t$. Empowered with the mask $M$ found by IB, our xDDPM method instead *only* denoise the features deemed relevant by $M$. Concretely, we introduce the following modified denoising objective:

$$L_{\epsilon_\theta, M} = ||\epsilon \odot M - \epsilon_\theta(\sqrt{\bar{\alpha}_t}x_0 + \sqrt{1 - \bar{\alpha}_t}\epsilon, t)||_2^2, \epsilon \sim \mathcal{N}(0, I) \tag{4}$$

Here $M \sim q_\varphi(M|X)$ is obtained by the VIB. For xDDPM, the above denoising objective and the VIB objective are jointly optimized, constituting the full xDDPM objective:

$$
\begin{aligned}
L_{\text{xDDPM}} &= L_{\epsilon_\theta, M} + \lambda \cdot L_{\text{VIB}} \\
&= \mathbb{E}\left[ ||\epsilon \odot M - \epsilon_\theta(\sqrt{\bar{\alpha}_t}X + \sqrt{1 - \bar{\alpha}_t}\epsilon, t)||_2^2 + \lambda \cdot \left( \log \frac{p_\varphi(X_S|X)}{q_\varphi(X_S)} - \beta \log q_\varphi(S|X_S) \right) \right]
\end{aligned}
\tag{5}
$$

Here the expectation is taken w.r.t. $(X, S) \sim p(X, S)$ (from data), $X_S \sim p_\varphi(X_S|X)$, and $\epsilon \sim \mathcal{N}(0, I)$. During training, the VIB objective $L_{\text{VIB}}$ helps discover the mask $M$ that indicates the relevant features of $X$ to $S$. For features that are found relevant, the corresponding mask elements in $M$ are near 1, so the xDDPM's denoising objective $L_{\epsilon_\theta, M}$ reverts to the DDPM objective $L_{\epsilon_\theta}$. On the other hand, for the features that are deemed irrelevant to $S$, the corresponding mask elements in $M$ are approximately 0, and the corresponding features in the noise target $\epsilon \odot M$ are 0. In other words, the $L_{\epsilon_\theta, M}$ trains $\epsilon_\theta$ to predict zero noise on these irrelevant features.

**Architecture design.** Here we detail the neural network architecture for xDDPM. We use grid-based data (e.g., image and videos) to illustrate the architectural design[4]. For the denoising network $\epsilon_\theta$, we

---

[4]Note that our method is fully general and can also deal with other types of input such as graph and sequence.

use the standard choice of U-Net Ronneberger et al. (2015). For modeling the distribution $p_\varphi(X_S|X)$, we first use a U-Net that takes as input $X$ and returns the estimation of mean $\mu_{M,\varphi}(X)$ and logit $\xi_{M,\varphi}(X)$ for the mask $M$ as its feature maps. Then we use the reparameterization trick Kingma & Welling (2013) to represent $p_\varphi(X_S|X)$ with the Gaussian $\mathcal{N}(X_S; \mu_{X_S,\varphi}(X), \sigma^2_{X_S,\varphi}(X))$, where the mean and the variance are conditioned on $X$ as follows:

$$\mu_{X_S,\varphi}(X) = \text{clamp}_{[0,1]}(\mu_{M,\varphi}(X)) \odot X$$
$$\sigma^2_{X_S,\varphi}(X) = \text{softplus}(\xi_{M,\varphi}(X) \odot X)$$

We employ $\text{clamp}_{[0,1]}$ to make sure that the mask mean stays between 0 and 1, and $\text{softplus}(x) = \log(1 + e^x)$ to ensure that the variance is non-negative. For the prior term $q_\varphi(X_S)$ in $L_{\text{VIB}}$ (Eq. 3), there are multiple options. The most general way is to use a mixture of full Gaussians where the mixing weight and the Gaussian mean and covariance matrix are learnable. Instead, we find that the simplest choice of letting $q_\varphi(X_S)$ be a diagonal Gaussian $\mathcal{N}(X_S; 0, I)$ works quite well. For $q_\varphi(S|X_S)$, we assume that $S$ obeys a diagonal Gaussian $\mathcal{N}(S; \mu_{S,\varphi}(X_S), I)$ where $\mu_{S,\varphi}(X_S)$ can be a U-Net encoder followed by a Multilayer Perceptron (MLP). Thus, $\log q_\varphi(S|X_S)$ reduces to a standard MSE loss. Combined together, $L_{\text{VIB}}$ in Eq. 3 reduces to

$$L_{\text{VIB}} = \mathbb{E}_{(X,S)\sim p(X,S)} \left[ \frac{1}{2}\beta ||\mu_{S,\varphi}(X_S) - S||^2_2 \right.$$
$$\left. + \frac{1}{2}\Big(1 + \sum_{j=1}^{D} \big( \log \sigma^2_{X_S,\varphi}(X) - \mu^2_{X_S,\varphi}(X) - \sigma^2_{X_S,\varphi}(X)\big)_j \Big) \right] \quad (6)$$

Here in the second term, the $j$ denotes the $j$th feature for the variable, and it sums over $D$ dimensions since $X_S \in R^D$. For detailed derivation, see Appendix A.8.

**Remark on learning speed.** Since xDDPM only needs to learn to denoise the relevant features which contain much less information than the full variable, it is much more efficient to train than the original DDPM. Empirically, we find that xDDPM typically converges around twice as fast as the DDPM method. This demonstrates the added benefits of how xDDPM for improving learning speed.

## 4 EXPERIMENTS

In this section, we aim to answer the following questions: (1) How is the explanation ability of our method xDDPM compared with other state-of-the-art methods (*e.g.*, SmoothGrad and GuidedGrad-CAM) on three datasets (wetting, tension and fluid) in terms of three different metrics (namely IoU, Sensitivity-n and Correlation)? (2) How does each component of our method xDDPM affect its explanation capability? (3) What is the efficiency of xDDPM compared with other baselines?

We provide results and experiment analysis in supplementary.

## 5 CONCLUSION

Existing diffusion models lack explainability for high-dimensional observations and accompanying signals. To address this, we introduce xDDPM, an Explainable Denoising Diffusion Probabilistic Model. By training the denoising network to focus on relevant components and leaving non-relevant parts noisy, xDDPM achieves explainable generation. Incorporating the Information Bottleneck aids in discovering the relevant components. Experimental results on cell dynamics and fluid dynamics datasets consistently demonstrate xDDPM's explainable generation ability. In Appendix A.12, we also discuss broader impacts and limitations of our method. In summary, xDDPM holds significant potential in bridging the gap between accurate generation and interpretability within diffusion models, critical for modeling tasks across science and engineering.

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

## A  APPENDIX

### A.1  RELATED WORK

**Denoising diffusion probabilistic models for Artificial Intelligence**. Denoising diffusion probabilistic models have demonstrated their capability to predict the dynamic evolution in various domains such as fluid dynamics Cachay et al. (2023), weather forecasting Price et al. (2023), and molecular dynamics Wu et al. (2022). They have also been effectively applied in inverse design tasks, enabling the optimization of airfoils Wu et al. (2024) and proteins Watson et al. (2023). Additionally, diffusion models have shown promise in solving complex inverse problems Holzschuh et al. (2023). These are just a few examples of the diverse applications where diffusion models have been successfully employed. In the realm of biology, researchers have utilized the DDPM to model diffusion processes in biological networks Fu et al. (2023); Best & Hummer (2011); Gao et al. (2023); Xu et al. (2022), enabling the analysis of protein-protein interactions and gene regulatory networks. In the field of physics, the DDPM has been applied to study the diffusion of particles in complex systems, such as the spread of heat in materials. Furthermore, in the domain of chemistry, the DDPM has been employed to understand the diffusion of molecules and reactions in chemical systems. These studies highlight the versatility and effectiveness of the DDPM in capturing and analyzing diffusion dynamics across various scientific disciplines Xu et al. (2022). Ongoing research aims to further explore its potential for solving complex problems in AI for Science.

**Explanation for Artificial Intelligence**. Explanation, also known as attribution discovery, is an ever-evolving research field that has seen the development of various methods aimed at understanding the significance and contribution of different input features. Gradient Baehrens et al. (2010) and Saliency Simonyan & Zisserman (2014) compute the gradient of the target output neuron in relation to the input features. SmoothGrad Smilkov et al. (2017) enhances gradient-based attribution maps by averaging gradients across multiple inputs, employing techniques such as brightness level interpolations or considering a local neighborhood. Another approach, Guided Backpropagation (GuidedBP) Springenberg et al. (2014), modifies the propagation rule. Grad-Cam Selvaraju et al. (2017) utilizes the activations of the final convolutional layer to calculate relevance scores. They also combine their method with GuidedBP, resulting in GuidedGrad-CAM Ribeiro et al. (2016). In a manner similar to our work, MacDonald et al. (2019) adopts a rate-distortion perspective; however, their focus is on minimizing the norm of the mask rather than emphasizing shared information. The goal of Schulz et al. (2020) is to incorporate the information bottleneck as an explanatory component to shed light on fixed and trained neural networks using standard stochastic gradient methods. These attribution techniques like Schulz et al. (2020) necessitate that the task at hand involves classification and their objective is to associate the decision made for a particular class with the significance of pixels in a provided dataset. As far as we know, our research is the first to estimate the amount of information utilized for attribution purposes in diffusion models applied to scientific modeling.

### A.2  PRELIMINARY FOR THE DDPM METHOD

The Denoising Diffusion Probabilistic Model (DDPM) Ho et al. (2020) consists of two essential processes: the forward process (or diffusion process) and the reverse process. Let's focus on describing the forward process first. The forward process in a diffusion model approximates the posterior distribution $q(x_{1:T}|x_0)$, which represents the sequence of latent variables $x_{1:T}$ given an initial value $x_0$. This approximation is achieved by iteratively applying a Markov chain that adds Gaussian noise gradually over time. Specifically, the forward process is represented as follows:

$$q(x_t|x_{t-1}) = \mathcal{N}\left(x_t; \mu_t(x_{t-1}), \beta_t \mathbf{I}\right) \tag{7}$$

Here, $x_t$ denotes the latent variable at time step $t$, and $x_{t-1}$ is the variable at the previous time step. The distribution $q(x_t|x_{t-1})$ is modeled as a Gaussian distribution with mean $\mu_t(x_{t-1})$ and variance $\beta_t \mathbf{I}$, where $\beta_t$ is the variance parameter at time step $t$, and $\mathbf{I}$ represents the identity matrix. The mean $\mu_t(x_{t-1})$ can depend on the previous latent variable $x_{t-1}$ and is typically modeled using neural networks or other parameterized functions. By sequentially applying the distribution $q(x_t|x_{t-1})$ for each time step, starting from the initial value $x_0$, we obtain an approximation of the posterior distribution $q(x_{1:T}|x_0)$ that captures the temporal evolution of the latent variables via $q(x_{1:T}|x_0) := \prod_{t=1}^{T} q(x_t|x_{t-1})$.

To describe the reverse process, let's consider a diffusion model with $T$ time steps. Given an observed data point $x_T$ at the final time step, the goal is to generate a sample from the initial distribution $p(x_0)$. The reverse process in a diffusion model can be formulated as follows: (1) Initialization: Set $x_T$ as the observed data point. (2) Iterative Sampling: Starting from $t = T - 1$ and moving backwards until $t = 0$, sample $x_t$ from the distribution $p(x_t|x_{t+1})$, where $p(x_t|x_{t+1})$ represents the reverse diffusion process.

The distribution $p(x_t|x_{t+1})$ in the reverse process is typically modeled as a Gaussian distribution, similar to the forward process. However, the mean and variance parameters are adjusted to account for the reverse direction. The specific form of $p(x_t|x_{t+1})$ is defined as follows:

$$p_\theta(x_{0:T}) := p(x_T) \prod_{t=1}^{T} p_\theta(x_{t-1}|x_t) \tag{8}$$

$$p(x_t|x_{t+1}) := \mathcal{N}(x_t; \mu_\theta(x_{t+1}, t + 1), \sum_\theta(x_{t+1}, t + 1))$$

By iteratively sampling from the reverse process, we can generate a sequence of latent variables $x_{0:T}$ that follows the reverse diffusion process. This reverse sequence represents a sample from the initial distribution $p(x_0)$. The reverse process is crucial for training the diffusion model. During training, the model learns to approximate the reverse process by minimizing the discrepancy between the generated samples and the observed data points. This training procedure ensures that the model captures the underlying data distribution and can generate realistic samples.

The optimization objective of the diffusion model is conducted via the following negative log likelihood:

$$\mathbb{E}[-\log p_\theta(x_0)] \leq \mathbb{E}_q[-\log \frac{p_\theta(x_{0:T})}{q(x_{1:T}|x_0)}]$$

$$= \mathbb{E}_q[-\log p(x_T) - \sum_{t \geq 1} \log \frac{p_\theta(x_{t-1}|x_t)}{q(x_t|x_{t-1})}] =: \mathcal{L} \tag{9}$$

## A.3 METHOD

**The DDPM method.** To tackle the above task, we build upon the recent advances of Denoising Diffusion Probablistic Models (DDPMs) Ho et al. (2020). DDPM is an elegant method to learn high-dimensional distributions $P_X(x)$ given data samples $\{x^{(n)}\}_{n=1}^{N}$. Concretely, DDPM consists of a forward process that adds $t$ steps of Gaussian noise to the data sample[5] $x_0$ to obtain a noisy sample $x_t$: $x_t = \sqrt{\bar{\alpha}_t}x_0 + \sqrt{1 - \bar{\alpha}_t}\epsilon$, where $\epsilon \sim \mathcal{N}(0, I)$ has the same dimension as $x_0$ and $\bar{\alpha}_t$ is a predefined schedule. Since the summation of Gaussian is also a Gaussian, the $t$ steps of adding Gaussian noise is equivalent to adding a single Gaussian $\sqrt{1 - \bar{\alpha}_t}\epsilon$. After $T$ steps of forward process, $x_T$ approximates a standard Gaussian distribution. In the reverse process, the DDPM learns a denoising model $\epsilon_\theta(x_t, t)$ that aims to revert the forward process:

$$x_{t-1} = \frac{1}{\sqrt{\alpha_t}} \left( x_t - \frac{\sqrt{1 - \alpha_t}}{\sqrt{1 - \bar{\alpha}_t}} \epsilon_\theta(x_t, t) \right) + \sigma_t \eta, t = T, ...1.$$

Here $\alpha_t, \sigma_t$ are pre-defined schedule and $\eta \sim \mathcal{N}(0, I)$. To learn $\epsilon_\theta(x_t, t)$, DDPM use the denoising objective

$$L_{\epsilon_\theta} = ||\epsilon - \epsilon_\theta(\sqrt{\bar{\alpha}_t}x_0 + \sqrt{1 - \bar{\alpha}_t}\epsilon, t)||_2^2, \epsilon \sim \mathcal{N}(0, I) \tag{10}$$

which aims to predict the added noise $\epsilon$ based on the noisy sample $x_t = \sqrt{\bar{\alpha}_t}x_0 + \sqrt{1 - \bar{\alpha}_t}\epsilon$. For more details about DDPM, see Appendix A.2.

**xDDPM method**. At inference time, xDDPM generates the explainable variable $W$ using the same procedure as DDPM:

$$x_{t-1} = \frac{1}{\sqrt{\alpha_t}} \left( x_t - \frac{\sqrt{1 - \alpha_t}}{\sqrt{1 - \bar{\alpha}_t}} \epsilon_\theta(x_t, t) \right) + \sigma_t \eta, t = T, ...1. \tag{11}$$

---

[5]Here the subscript $t$ in $x_t$ denotes the denoising step.

---

**Algorithm 1** xDDPM training

---

1: **repeat**
2: $x_0 \sim p(x_0)$
3: $\epsilon \sim \mathcal{N}(0, I)$
4: Take gradient descent step of $L_{\text{xDDPM}}$ w.r.t. $\theta$ and $\varphi$:
  $\nabla_{(\theta,\varphi)}[||\epsilon \odot M - \epsilon_\theta(\sqrt{\bar{\alpha}_t}x_0 + \sqrt{1 - \bar{\alpha}_t}\epsilon, t)||_2^2 + \lambda L_{\text{VIB}}]$
5: **until** converged

---

starting with $x_T \sim \mathcal{N}(0, I)$, and we have $W := x_0$ as the last sample. We see that at inference, the irrelevant features are not denoised and remain Gaussian because the $\epsilon_\theta$ is trained to predict 0 for these features. In this way, we can generate the explainable variable $W$ in inference time where only the relevant features of $S$ are denoised.

Taken together, the modified denoising objective $L_{\epsilon_\theta,M}$ (Eq. 4) and the IB module for discovering the relevant features (Eq. 3) constitute the key innovation of xDDPM. We provide the algorithm for training as Alg. 1.

**Architecture design.** Here we detail the neural network architecture for xDDPM. We use grid-based data (e.g., image and videos) to illustrate the architectural design[6]. For the denoising network $\epsilon_\theta$, we use the standard choice of U-Net Ronneberger et al. (2015). For modeling the distribution $p_\varphi(X_S|X)$, we first use a U-Net that takes as input $X$ and returns the estimation of mean $\mu_{M,\varphi}(X)$ and logit $\xi_{M,\varphi}(X)$ for the mask $M$ as its feature maps. Then we use the reparameterization trick Kingma & Welling (2013) to represent $p_\varphi(X_S|X)$ with the Gaussian $\mathcal{N}(X_S; \mu_{X_S,\varphi}(X), \sigma^2_{X_S,\varphi}(X))$, where the mean and the variance are conditioned on $X$ as follows:

$$\mu_{X_S,\varphi}(X) = \text{clamp}_{[0,1]}(\mu_{M,\varphi}(X)) \odot X$$

$$\sigma^2_{X_S,\varphi}(X) = \text{softplus}(\xi_{M,\varphi}(X) \odot X)$$

We employ $\text{clamp}_{[0,1]}$ to make sure that the mask mean stays between 0 and 1, and $\text{softplus}(x) = \log(1 + e^x)$ to ensure that the variance is non-negative. For the prior term $q_\varphi(X_S)$ in $L_{\text{VIB}}$ (Eq. 3), there are multiple options. The most general way is to use a mixture of full Gaussians where the mixing weight and the Gaussian mean and covariance matrix are learnable. Instead, we find that the simplest choice of letting $q_\varphi(X_S)$ be a diagonal Gaussian $\mathcal{N}(X_S; 0, I)$ works quite well. For $q_\varphi(S|X_S)$, we assume that $S$ obeys a diagonal Gaussian $\mathcal{N}(S; \mu_{S,\varphi}(X_S), I)$ where $\mu_{S,\varphi}(X_S)$ can be a U-Net encoder followed by a Multilayer Perceptron (MLP). Thus, $\log q_\varphi(S|X_S)$ reduces to a standard MSE loss. Combined together, $L_{\text{VIB}}$ in Eq. 3 reduces to

$$L_{\text{VIB}} = \mathbb{E}_{(X,S)\sim p(X,S)} \left[ \frac{1}{2}\beta||\mu_{S,\varphi}(X_S) - S||_2^2 \right.$$

$$\left. + \frac{1}{2}\left(1 + \sum_{j=1}^{D}\left(\log \sigma^2_{X_S,\varphi}(X) - \mu^2_{X_S,\varphi}(X) - \sigma^2_{X_S,\varphi}(X)\right)_j\right) \right] \tag{12}$$

Here in the second term, the $j$ denotes the $j$th feature for the variable, and it sums over $D$ dimensions since $X_S \in R^D$. For detailed derivation, see Appendix A.8.

**Remark on learning speed.** Since xDDPM only needs to learn to denoise the relevant features which contain much less information than the full variable, it is much more efficient to train than the original DDPM. Empirically, we find that xDDPM typically converges around twice as fast as the DDPM method. This demonstrates the added benefits of how xDDPM for improving learning speed.

## A.4 EXPERIMENTS

**Datasets.** Our datasets include two cell dynamics datasets and a fluid dynamics dataset. Detailed information about these datasets, including specific statistics, can be found in Table 3 in Appendix A.10. The wetting dataset explores morphological changes of a system (such as cells) under various scenarios involving grid and domain configurations, tension, adhesion, and other physical parameters. It

---

[6]Note that our method is fully general and can also deal with other types of input such as graph and sequence.

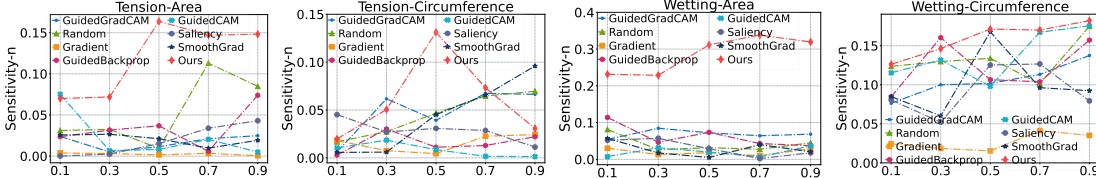

Figure 2: The Sensitivity-n values of all methods were evaluated on two datasets, specifically tension and wetting. The range of masked pixels varied from 10% to 90%. The definition of Sensitivity-n can be found in Eq. 14. It was observed that the Sensitivity-n values did not significantly differentiate between the baseline methods. However, our proposed method xDDPM outperformed the others for most masking percentages, demonstrating that our method can discover relevant pixels that are highly correlated to the prediction of the signal.

is generated through simulations with systematically modified key parameters to observe wetting behavior. In contrast, the tension dataset examines how the shape of a system evolves under specific grid and domain conditions, achieved by adjusting parameters like tension parameters. As an example, in Figure 3 (b), each trajectory $X$ consists of cell states across 20 time steps. We see that the cell shape gradually becomes more rounded as time progresses within each trajectory due to the surface tension. In addition, we have accompanying signal $S$ of cell areas or circumferences. Our focus extends beyond learning a probabilistic model for generating cell trajectory videos. We are equally interested in identifying the which video pixels corresponds to the signal of area or circumferences. For the Fluid dataset, please refer to Appendix A.9 for a detailed explanation. As exemplified in Figure ?? (b), our interest lies not only in predicting the pressure field conditioned on the boundary but also in discerning which components of the pressure field contribute to the force measurements acting on the boundary.

**Evaluation Metrics.** Building upon previous research Schulz et al. (2020); Ancona et al. (2017); Ahmadzadeh et al. (2021), we have adopted Intersection over Union (IoU) and Sensitivity-n as the evaluation metrics for all the methods in our study. These metrics are widely used in the field and provide valuable insights into the performance of the approaches. The definition of IoU is shown as follows:

$$IoU = \frac{Area\ of\ Overlap}{Area\ of\ Union} \tag{13}$$

IoU is a value between 0 and 1, where a higher value indicates a better overlap between the predicted and ground truth regions. The rationale behind utilizing IoU is to calculate the intersection over union between the generated images and the ground-truth cases, considering the corresponding signal variances such as cell areas or circumferences.

The Sensitivity-n metric, introduced by Ancona et al. (2017), evaluates attribution methods by randomly masking network inputs and quantifying the correlation between the masked attribution and the corresponding decrease in classifier score. For regression task, Sensitivity-n computes the Pearson correlation coefficient between the attribution values and the associated deviation in prediction when masking a set $M_n$ of $n$ randomly selected pixel indices:

$$\text{Sensitivity-n} = \text{corr}\left(\sum_{i \in M_n} R_i(x), \left(S(x) - S(x_{[x_{M_n}=0]})\right)^2\right) \tag{14}$$

Here, $R_i$ represents the relevance of the indexed pixel $i$, $S(x)$ denotes the prediction from the decode network with clean inputs $x$, and $S(x_{[x_{M_n}=0]})$ represents the prediction value from the decode network after randomly setting the masked pixels to 0. Intuitively, by masking random pixels, the prediction $S(x_{[x_{M_n}=0]})$ will deviate from the prediction $S(x)$ that is obtained without masking the pixels. Sensitivity-n quantifies how the reduction of the prediction performance $(S(x) - S(x_{[x_{M_n}=0]}))^2$ is correlated with the total relevance $\sum_{i \in M_n} R_i(x)$ a method assigns to the masked pixels. A good attribution method should assign high relevance values to the pixels such that when those pixels are masked, the prediction performance drop significantly, producing a high Sensitivity-n value.

**Baselines.** We performed comprehensive experiments, comparing our method xDDPM with several state-of-the-art baselines on three distinct datasets. These baselines include Random Schulz et al.

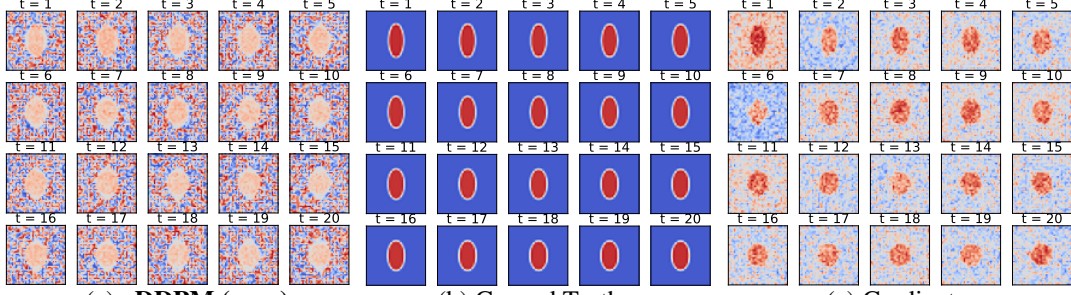

Figure 3: We present visualization results comparing our proposed method, xDDPM, with the best baseline, Gradient, on the Tension dataset, specifically in terms of cells' area. The middle section represents the ground-truth case, while the left and right parts exhibit the generated results of 20 time steps using xDDPM and Gradient, respectively. The mixed colors indicate the irrelevant portion to the signal variable (cells' area), while the more smooth, uniform color represents relevant parts. Methods that generate cells more similar to the ground truth exhibit higher explanatory abilities. In this regard, xDDPM demonstrates superior explanatory ability as the generated images at each time step closely resemble those of the ground truth.

(2020), Gradient Baehrens et al. (2010), Saliency Simonyan & Zisserman (2014), GuidedBP Springenberg et al. (2014), GuidedCAM Selvaraju et al. (2017), SmoothGrad Smilkov et al. (2017), and GuidedGrad Ribeiro et al. (2016). To help reproduce the experiment results of all methods, we also provide detailed implementations of all baselines in Appendix A.10.

## A.5   OVERALL EXPLANATION ABILITY COMPARISON

We evaluate all methods using three metrics: IoU, Sensitivity-n (50% pixel masking), and Correlation (Table 1). Visual results of state-of-the-art baselines and our proposed xDDPM are presented in Figures 3 and **??**. Additional experiments involve masking pixels in generated samples to assess explanation capability (Figure 2). Sensitivity-n average values are provided in Table 4 (Appendix A.10). From these results, we make the following observations.

**Results**. Based on the results presented in Table 1, we observed that our method consistently outperforms all other approaches across all metrics on the three datasets. This can be attributed to several key factors. Firstly, our method xDDPM employs the Information Bottleneck mechanism to guide the discovery of relevant features related to the signal variable. By emphasizing the generation of relevant parts in the samples, our method xDDPM effectively improves the relevance of the generated samples to the cell areas, circumferences, and pressures of the fluid dataset. This is achieved by maximizing the mutual information between the diffused samples and the signal variable. In comparison, the baselines generate samples without such emphasis. Secondly, our proposed mask mechanism, applied to the Gaussian noise, plays a crucial role in generating samples that closely resemble the ground truth samples. Furthermore, we observe that Gradient, Saliency, and SmoothGrad outperform the other methods on all three datasets. These baselines leverage gradient-based techniques to capture the relevant parts related to the signal variable successfully.

In Figure 2, we present the Sensitivity-n values obtained by all methods on images from the wetting and tension datasets. We see that our method, xDDPM, gives a much higher Sensitivity-n values than the other baselines in most conditions (masking percentage). This shows that our method has the strongest ability to identify relevant pixels that are highly correlated to the prediction of the signal.

**Visualization comparison**. Figures 3 and **??** display generated samples from the best baseline, Gradient, and our proposed method. In Figure 3, mixed colors denote *non-relevant* parts to the signal (cell areas, circumference, and pressures) as identified by the methods. We see that: (1) Both our method and Gradient can generate samples with relevant parts to cell areas, but our method achieves even higher relevance and explanation ability. (2) In Figure **??**, our method correctly discovers the relevant parts of pressure field (the front of the jellyfish) that contribute to the force on the boundary, while the best baseline fails to generate relevant parts. This demonstrates the effectiveness and general applicability of our method.

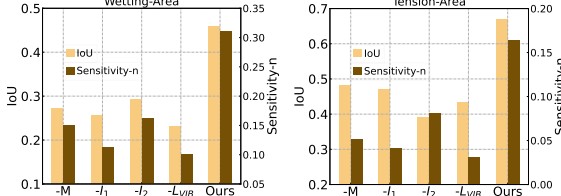

Figure 4: Ablation study of **xDDPM** on the wetting and tension datasets. We considered four variants for our method, which are as follows: (1) "-$M$ which removes the continuous mask used for Gaussian noise; (2) "-$I_1$ (representing $I(X_S; S)$)" which eliminates the mutual information between the representations of data samples and the signal variable; (3) "-$I_2$ (representing $I(X; X_S)$)" which removes the mutual information between the data samples and their corresponding representations; (4) "-$L_{\text{VIB}}$" which excludes the Information Bottleneck mechanism that guides our method xDDPM. We notice that each component of xDDPM contributes the final performance.

Table 2: Training time (in hours) is reported for each method on three datasets. Our proposed method, xDDPM, demonstrates faster training times compared to most methods, except for Random, which achieves the best performance in terms of IoU.

| Methods | Tension | | Wetting | | Fluid | |
|---|---|---|---|---|---|---|
| | Time (h) | IoU | Time (h) | IoU | Time (h) | Correlation |
| Random | 2.16 | 0.2398 | 2.13 | 0.3297 | 2.21 | 0.0003 |
| Gradient | 4.83 | 0.2147 | 5.33 | 0.5309 | 5.33 | 0.0064 |
| Saliency | 5.18 | 0.2811 | 4.83 | 0.4896 | 5.33 | 0.0327 |
| GuidedBP | 5.33 | 0.1033 | 5.33 | 0.30350 | 5.25 | 0.0033 |
| GuidedCAM | 5.16 | 0.0601 | 5.16 | 0.0399 | 6.25 | 0.0085 |
| SmoothGrad | 11.08 | 0.1755 | 10.33 | 0.4691 | 12.16 | 0.0093 |
| GuidedGrad | 8.55 | 0.0723 | 8.67 | 0.1272 | 9.15 | 0.0285 |
| **xDDPM (ours)** | 2.35 | **0.4590** | 2.27 | **0.6689** | 8.19 | **0.3343** |

## A.6 ABLATION STUDY

We conducted experiments to assess the impact of each component in our method, presented in Figure 4. The four variants considered are: (1) "-$M$" (eliminates the continuous mask for Gaussian noise), (2) "-$I_1$ (rep. $I(X_S; S)$)" (removes mutual information between data sample representations and the signal variable), (3) "-$I_2$ (rep. $I(X; X_S)$)" (eliminates mutual information between data samples and their representations), and (4) "-$L_{\text{VIB}}$" (excludes the Information Bottleneck mechanism guiding our method). By comparing these variants with the original xDDPM, we evaluate the significance of each component in improving performance. From the analysis in Figure 4, we observe that all the components of our method are essential to its performance, further validating the effectiveness of our approach. Additionally, we note that the Information Bottleneck (IB) mechanism (-$L_{\text{VIB}}$) demonstrates the most substantial impact on the effectiveness of our method.

## A.7 EFFICIENCY COMPARISON

In this section, we examine the training time required for all methods and present the results in Table 2. To ensure fairness, all methods are implemented in Python 3.9, PyTorch 2.1 (GPU version). The experiments are conducted on a server equipped with 48 cores and 8 Nvidia GeForce H800 GPUs. Our observations reveal that our method, xDDPM, achieves competitive model efficiency compared to the baselines across the three datasets with outstanding performance. This validates that our method xDDPM is well-suited for application on large-scale datasets, while maintaining a stable explanatory ability. Among the baselines, SmoothGrad stands out as the slowest due to its approach of enhancing gradient-based attribution maps by averaging gradients across multiple inputs, albeit with competitive performance.

## A.8 Derivation of empirical VIB

In this section, we provide the deduction progress for the following formula:

$$L_{\text{VIB}} = \mathbb{E}_{(X,S)\sim p(X,S)} \left[ \frac{1}{2} \big( 1 + \sum_{j=1}^{D} \big( \log \sigma_{X_S,\varphi}^2(X) - \mu_{X_S,\varphi}^2(X) - \sigma_{X_S,\varphi}^2(X) \big)_j \big) + \frac{1}{2}\beta \|\mu_{S,\varphi}(X_S) - S\|_2^2 \right]$$

(15)

Assuming that the prior $q_\varphi(X_S)$ and the posterior approximation $p_\varphi(X_S|X)$ are both Gaussian distributions, we proceed with the following notation. Let $D$ represent the dimensionality of $X_S$. Through reparameterization trick, the variational mean and standard deviation evaluated on $X$ are $\mu_{X_S,\varphi}(X)$ and $\sigma_{X_S,\varphi}^2(X)$, respectively. Consequently, we have the following relationship:

$$\int q_\varphi(X_S) \log p(X_S) dX_S = \int \mathcal{N}(X_S; \mu_{X_S,\varphi}(X), \sigma_{X_S,\varphi}^2(X)) \log \mathcal{N}(X_S; \mathbf{0}, \mathbf{I}) dX_S \qquad (16)$$

$$= -\frac{D}{2}\log(2\pi) - \frac{1}{2}\sum_{j=1}^{D}(\mu_{X_S,\varphi}^2(X) + \sigma_{X_S,\varphi}^2(X))_j \qquad (17)$$

And:

$$\int q_\varphi(X_S) \log q_\varphi(X_S) dX_S = \int \mathcal{N}(X_S; \mu_{X_S,\varphi}(X), \sigma_{X_S,\varphi}^2(X)) \log \mathcal{N}(X_S; \mu_{X_S,\varphi}(X), \sigma_{X_S,\varphi}^2(X)) dX_S$$

(18)

$$= -\frac{D}{2}\log(2\pi) - \frac{1}{2}\sum_{j=1}^{D}(1 + \log \sigma_{X_S,\varphi}^2(X))_j \qquad (19)$$

Therefore, we can have the following deduction result:

$$-D_{\text{KL}}(q_\varphi(X_S)\|p_\varphi(X_S)) = \int q_\varphi(X_S)(\log p_\varphi(X_S) - q_\varphi(X_S))dX_S \qquad (20)$$

$$= \frac{1}{2}\big( 1 + \sum_{j=1}^{D} \big( \log \sigma_{X_S,\varphi}^2(X) - \mu_{X_S,\varphi}^2(X) - \sigma_{X_S,\varphi}^2(X) \big)_j \big) \qquad (21)$$

Thus, $\beta \log q_\varphi(S|X_S) = \beta \log(\mathcal{N}(S; \mu_{S,\varphi}(X_S), I)$ is deduced to a standard MSE loss, Thus, we can obtain the loss $L_{\text{VIB}} = \mathbb{E}_{(X,S)\sim p(X,S)} \left[ \frac{1}{2}\big( 1 + \sum_{j=1}^{D} \big( \log \sigma_{X_S,\varphi}^2(X) - \mu_{X_S,\varphi}^2(X) - \sigma_{X_S,\varphi}^2(X) \big)_j \big) + \frac{1}{2}\beta \|\mu_{S,\varphi}(X_S) - S\|_2^2 \right]$.

## A.9 Fluid Data Generation

We employ the Lily-Pad simulator Weymouth (2015) for generating both the training and testing datasets. The resolution of the 2D flow field is configured to be $128 \times 128$. It's noteworthy that in the context of Lily-Pad, the flow field is assumed to extend infinitely. The head of the jellyfish remains fixed at the coordinates $(25.6, 64)$. The representation of its two wings takes the form of identical ellipses, characterized by a fixed ratio of 0.15 between the shorter and longer axes. At every instant, symmetry is maintained across the central horizontal line defined by $y = 64$. To delineate the wing boundaries, we meticulously sample a total of $M = 20$ points along each wing. The pivotal parameter governing the jellyfish's control in this 2D experiment is the opening angle of the wings. This angle is defined as the deviation between the longer axis of the upper wing and the horizontal line. It serves as the crucial control signal, denoted as $w$.

Each trajectory originates with the widest possible opening angle and proceeds along a periodic cosine curve with a period of $T' = 200$. Trajectories are distinguished by variations in their

initial angle, angle amplitude, and phase ratio, denoted as $\tau$ – which represents the ratio between the closing duration and the entire pitching duration. For each trajectory, the initial angle $w_0$ is generated through a two-step process. Initially, a random mean angle $w^{(m)}$ is sampled within the range of $[20°, 40°]$. Subsequently, a random angle amplitude $w^{(a)}$ is sampled within the interval $[10°, \min(w^{(m)}, 60° - w^{(m)})]$. The resultant initial angle $w_0$ is then computed as $w_0 = w^{(m)} + w^{(a)}$, constrained within the range of $[10°, 60°]$. Meanwhile, the phase ratio $\tau$ is randomly chosen from the range of $[0.2, 0.8]$. The opening angle $w_t$ at step $t$ follows a specific pattern: it decreases from $w^{(m)} + w^{(a)}$ to $w^{(m)} - w^{(a)}$ as $t$ advances from 0 to $\tau T'$, and then it increases from $w^{(m)} - w^{(a)}$ to $w^{(m)} + w^{(a)}$ as $t$ progresses from $\tau T'$ to $T'$. Beyond this point, $w_t$ exhibits periodic variations for $t > T'$. This configuration aligns with previous studies on the propulsive performance of jellyfish Kang et al. (2023). Each trajectory is simulated for a total of 600 simulation steps, equivalent to 3 periods. To conserve space, only the segment of the trajectory spanning from $T' = 200$ to $3T' = 600$ steps is saved, with a step size of 10. This decision is made as the simulation from $t = 0$ to $T' = 200$ is primarily intended for initializing the flow field. Consequently, each trajectory is stored as a sequence comprising $\tilde{T} = (600 - 200)/10 = 40$ discrete steps.

In addition to tracking the positions of the wing boundary points and the opening angles $w$, we incorporate an image-like representation of the wing boundaries. This alternative representation contains valuable spatial information that can be more efficiently assimilated alongside the PDE states (fluid field) through convolutional neural networks. For each trajectory, this image-like boundary representation aligns seamlessly with the shape of the PDE states. At each time step, the boundaries of the two wings are combined and transformed into a tensor of dimensions [3, 64, 64]. Within each grid cell of this tensor, three distinct features are included: a binary mask indicating whether the cell resides within a boundary (marked as 1) or within the fluid (denoted as 0), and a relative position $(\Delta x, \Delta y)$ representing the distance from the cell center to the nearest point on the boundary. This representation enhances the compatibility between boundary information and PDE states. For every trajectory, we retain data on PDE states, opening angles, boundary points, boundary masks, offsets, and force data. These components are specified as follows:

- PDE states $u$: These have a shape of $[\tilde{T}, 3, 64, 64]$, representing the fluid field states for each time step, including velocity in the $x$ and $y$ directions and pressure. To conserve space, we downsample the resolution from $128 \times 128$ to $64 \times 64$.
  - velocity: $[\tilde{T}, 2, 64, 64]$.
  - pressure: $[\tilde{T}, 1, 64, 64]$.
- opening angels $w$: they have a shape of $[\tilde{T}]$. For each step, we save the opening angle in radians.
- boundary points: shape $[\tilde{T}, 2, M, 2]$. With a shape of $[\tilde{T}, 2, M, 2]$, we record the boundary points for both the upper and lower wings. Each wing comprises $M = 20$ points, and each point has 2 coordinates. To ensure compatibility with the downsampling of states, the coordinates in the $x$ and $y$ directions are scaled down to half ($64/128$) of their original values.
- boundary mask and offsets $b$: They have a shape of $[\tilde{T}, 3, 64, 64]$. For each time step, this includes a mask indicating the merged wings along with the half coordinates of boundary points and offsets in both the $x$ and $y$ directions. The resolution is $64 \times 64$, matching that of the PDE states.
  - mask: $[\tilde{T}, 1, 64, 64]$.
  - offsets: $[\tilde{T}, 2, 64, 64]$.
- force: it has a shape of $[\tilde{T}, 2]$. For each step, the simulator outputs the horizontal and vertical force from fluid to the jellyfish. The horizontal force is regarded as a thrust to jellyfish if positive and a drag otherwise.

We generated a total of $n = 45,000$ trajectories, each distinguished by varying parameters such as $w^{(a)}, w^{(m)}$, and $\tau$. Each trajectory occupies approximately 2MB of storage space, contributing to an overall dataset size of around 100GB. To create training samples, we employed sliding time windows that encompassed $T = 20$ consecutive time steps of both states and boundaries. This configuration

corresponds to $T' = 200$ original simulation steps, constituting a complete wing movement period. Consequently, each trajectory could generate up to 20 individual samples, resulting in a grand total of 6 million training samples. In each training sample, the opening angle remained consistent between the initial and final time steps due to the periodic nature of the motion. This consistency served as the control condition for our experiments. For test trajectories, we carefully selected the opening angle of the jellyfish at the initial time and used the initial states as the control conditions for both the initial and final time and state configurations.

## A.10 DETAILS OF BASELINES' IMPLEMENTATIONS

To help reproduce the experiment results of all methods, we also provide detailed implementations of all baselines in Appendix A.10. With the assistance of the mask M obtained through the regression function $\mu_{S,\varphi}$, the methods of our baselines focus on denoising only the features that are considered relevant according to the mask M.

$$L_{\epsilon_\theta, M} = ||\epsilon \odot M - \epsilon_\theta(\sqrt{\bar{\alpha}_t}x_0 + \sqrt{1 - \bar{\alpha}_t}\epsilon, t)||_2^2, \epsilon \sim \mathcal{N}(0, I) \tag{22}$$

$$L_{\text{Reg}} = \mathbb{E}_{(X,S) \sim p(X,S)}\left[||\mu_{S,\varphi}(X_S) - S||_2^2\right]$$

Combined together

$$L_{\text{baseline}} = L_{\epsilon_\theta, M} + \frac{1}{2}L_{\text{Reg}} \tag{23}$$

We employed seven distinct methods for obtaining masks:

- Random: Random involves utilizing random numbers as masks.
- Gradient: The gradient of $x_t$ is employed as the input to the model, and the gradient of $x_t$ is obtained through backpropagation, with the model adopting the architecture of Unet $U_\varphi$.

$$M = \frac{\partial U_\varphi(x_t)}{\partial x_t} \tag{24}$$

- Guided Backpropagation: Guided Backpropagation is a technique used in interpretability of neural networks. It enhances the visualization of feature importance by guiding the backpropagation process through the positive gradient values and setting negative gradient values to zero. The formulation for Guided Backpropagation is as follows:

$$\frac{\partial U_\varphi(x_t)}{\partial x_t} > 0 \Rightarrow M^+(x_t) = \frac{\partial U_\varphi(x_t)}{\partial x_t} \tag{25}$$

  For negative gradients:

$$\frac{\partial U_\varphi(x_t)}{\partial x_t} < 0 \Rightarrow M^-(x_t) = 0 \tag{26}$$

- SmoothGrad is a technique used to reduce noise in the interpretation of deep neural network predictions by averaging gradients over multiple perturbed instances of the input. The formulation for SmoothGrad is as follows:

$$M = \frac{1}{N}\sum_{j=1}^{N}\frac{\partial U_\varphi(\text{Perturb}(x_t))}{\partial x_{i,t}} \tag{27}$$

  $Perturb(x)$ is a function that introduces random noise or perturbation to the input $x$.

- GuidedCAM: Guided Class Activation Mapping (GuidedCAM) is a technique used for visualizing and interpreting the decisions of a convolutional neural network (CNN). It is often applied to understand the importance of different regions in an input image for a particular class prediction. The formulation for GuidedCAM is as follows:

$$M = ReLU(\frac{\partial U_\varphi(x_t)}{\partial x_t} \odot U_\varphi(x_t)) \tag{28}$$

Table 3: Data Statistics

| Datasets | Tension | | Wetting | | Fluid | |
|---|---|---|---|---|---|---|
| Types | Images | Videos | Images | Videos | Images | Videos |
| Number | 50,000 | 2,500 | 50,000 | 2,500 | 900,000 | 45,000 |

Table 4: The Sensitivity-n values of all methods were assessed on two datasets, specifically tension and wetting. The range of masked pixels varied from 10% to 90%. The definition of Sensitivity-n can be found in Equation 10. It was observed that the Sensitivity-n values did not exhibit significant differentiation among the baseline methods, as depicted in Figure 2. We have averaged the results shown in Figure 2 in this table. However, our proposed method xDDPM consistently outperformed the others for most masking percentages. This demonstrates that our method can effectively identify relevant pixels that have a strong correlation with the signal's prediction.

| Datasets | Wetting | | Tension | | Jellyfish |
|---|---|---|---|---|---|
| - | Area | Circumference | Area | Circumference | Pressure |
| Method | Sensitivity-n | Sensitivity-n | Sensitivity-n | Sensitivity-n | Sensitivity-n |
| Random | 0.0423 | 0.1326 | 0.0541 | 0.0450 | 0.1992 |
| Gradient | 0.0200 | 0.0269 | 0.0024 | 0.0154 | 0.0107 |
| Saliency | 0.0315 | 0.0927 | 0.0189 | 0.0286 | 0.1653 |
| GuidedBP | 0.0627 | 0.1224 | 0.0343 | 0.0158 | 0.1783 |
| GuidedCAM | 0.0203 | 0.1375 | 0.0230 | 0.0081 | 0.1804 |
| SmoothGrad | 0.0279 | 0.1003 | 0.0204 | 0.0436 | 0.0132 |
| GuidedGrad-CAM | 0.0687 | 0.1057 | 0.0172 | 0.0482 | 0.1953 |
| **xDDPM (ours)** | **0.2857** | **0.1568** | **0.1201** | **0.0610** | **0.3181** |

- Guided Grad-CAM (Guided Gradient-weighted Class Activation Mapping) combines the concepts of Guided Backpropagation and Grad-CAM to highlight important regions in an input image for a specific class prediction. The formulation for GuidedGradCAM is as follows:

$$M = ReLU(\frac{\partial U_\varphi(x_t)}{\partial x_t} \odot U_\varphi(x_t)) \odot ReLU(\frac{\partial U_\theta(x_t)}{\partial x_t} \odot U_\theta(x_t)) \quad (29)$$

- Saliency is an attribution method used to understand the importance of input features for a given model prediction. The formulation for Integrated Gradients is as follows:

$$M = (x_t - x_t^{'}) \odot \int_{\alpha=0}^{1} \frac{\partial U_\varphi(Interpolate(x^{'}, x, \alpha)}{\partial x_t} d\alpha \quad (30)$$

$Interpolate(x^{'}, x, \alpha)$ is a function that linearly interpolates between the baseline $x_t^{'}$ and the actual input $x_t$ at a given scale $\alpha$. The baseline $x_t^{'}$ is the mean of $x_t$

Considering all of the aforementioned baselines, we have constrained the range of M to be clamped within the interval $[0.7, 1.0]$. This adjustment aids the baselines in achieving improved performance across all datasets.

## A.11 TRAINING DETAILS AND PARAMETER SETTINGS OF XDDPM

To ensure fairness and consistency in our experiments, we made several design choices regarding the configuration of our model. The hidden dimension of the U-Net neural network was set to 64, providing an appropriate level of complexity for the task at hand. To balance the contribution of the Information Bottleneck loss ($L_{\text{VIB}}$), we assigned a weight $\lambda$ of 0.1 in Eq. 5. Additionally, we set the value of $\beta$ to 1 for all datasets, promoting a suitable trade-off between reconstruction accuracy and information preservation. For the Gaussian diffusion process, we performed 1000 diffusion steps to allow for comprehensive information exchange. The channel size multiplier of the U-Net neural networks was set to $[1, 2, 4, 8]$, ensuring effective feature extraction across various scales. The number of channels for the two cell dynamics datasets and the fluid dataset was set to 20 and 42, respectively, accommodating the unique characteristics of each dataset. Three datasets were standardized to an image size of $32 \times 32$ pixels. To facilitate convergence, we adopted a learning rate of $8 \times 10^{-5}$.

## A.12 BROADER IMPACTS AND LIMITATIONS

Our approach, xDDPM, expands the implementation of explanatory capabilities in denoising diffusion probabilistic models within the AI for science domain. This advancement enables efficient utilization

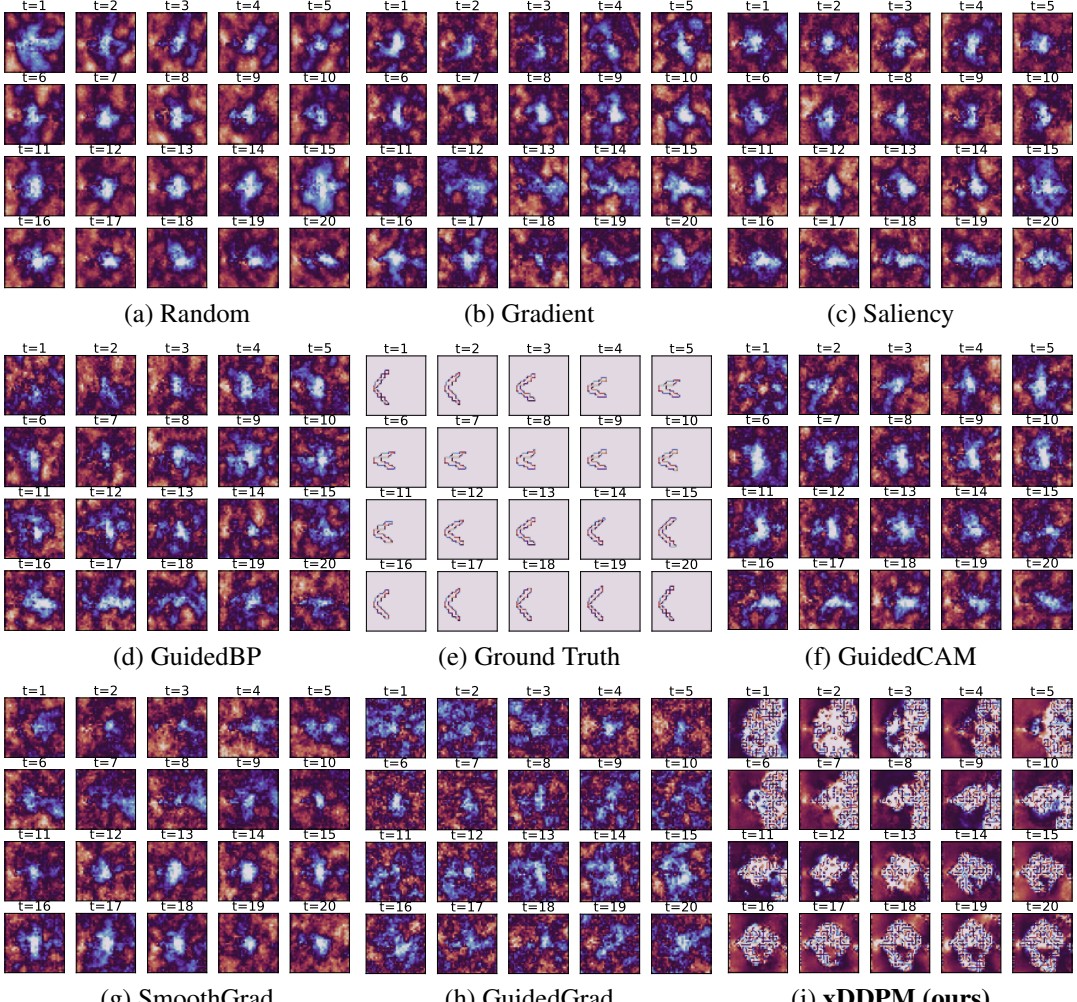

Figure 5: We present visualizations comparing our proposed method, xDDPM, with all baselines, on the Fluid dataset. The central subfigure represents the actual boundary of a jelly-like robot, which serves as a condition for generating the fluid pressure fields, denoted as X. The other subfigures depict the generated pressure field X by our xDDPM and other baselines. The blended colors indicate the irrelevant parts of the pressure fields to the force measurement signal S, as identified by the methods, while the smoother colors represent the relevant parts. We observe that our xDDPM accurately identifies that the enclosing region of the pressure fields is most relevant to the force measurement on the boundary. Furthermore, the generated pressure field exhibits consistent angles with respect to the given boundary. In contrast, the other baselines fail to generate a consistent pressure field and struggle to identify the relevant pixels associated with the signal.

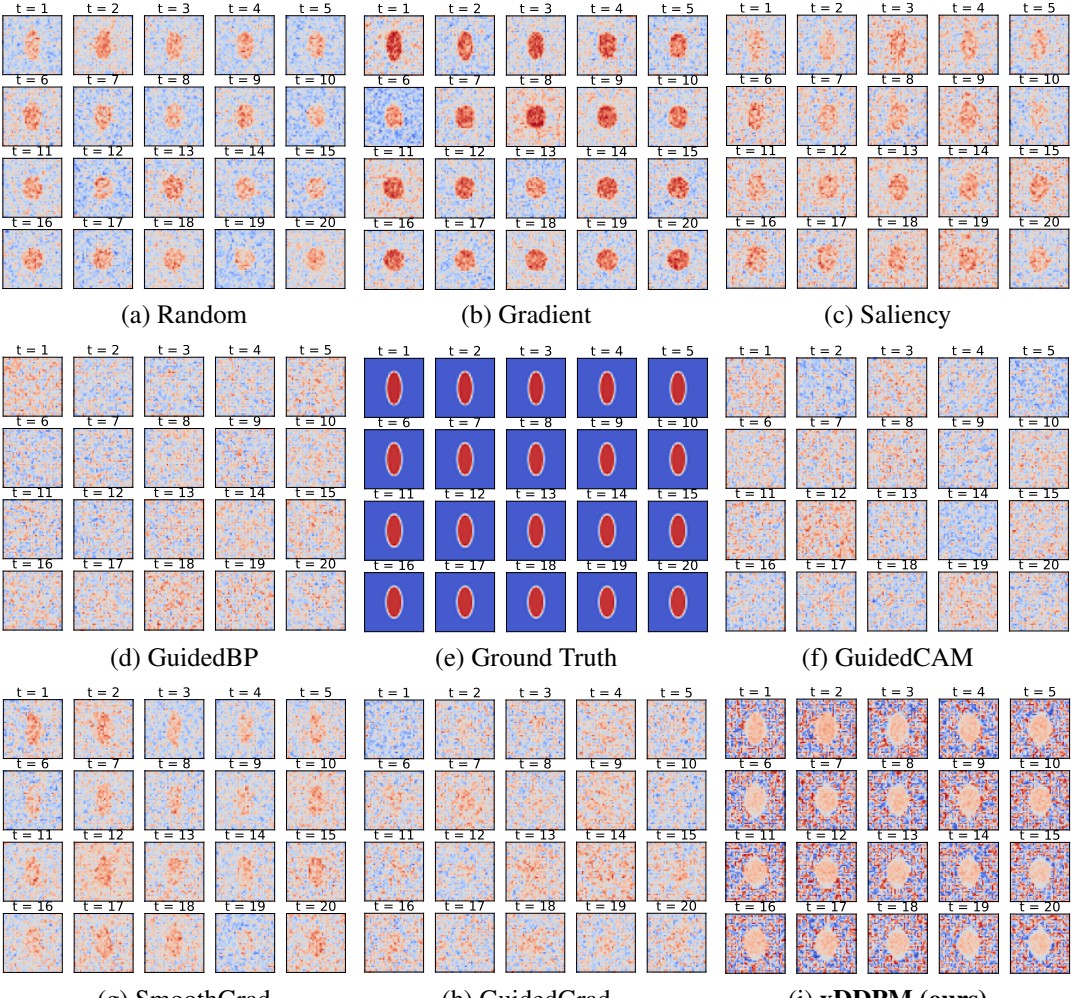

Figure 6: We showcase visualization results that compare our proposed method, xDDPM, with all baselines on the Tension dataset, specifically focusing on cells' area. The central portion represents the ground truth scenario, while the other figures display the generated results of 20 time steps using xDDPM and all baselines, respectively. The mixed colors indicate the irrelevant portion to the signal variable (cells' area), while the smoother and more uniform color represents the relevant parts. Methods that generate cells more similar to the ground truth demonstrate higher explanatory capabilities. In this regard, xDDPM exhibits superior explanatory ability as the generated images at each time step closely resemble those of the ground truth.

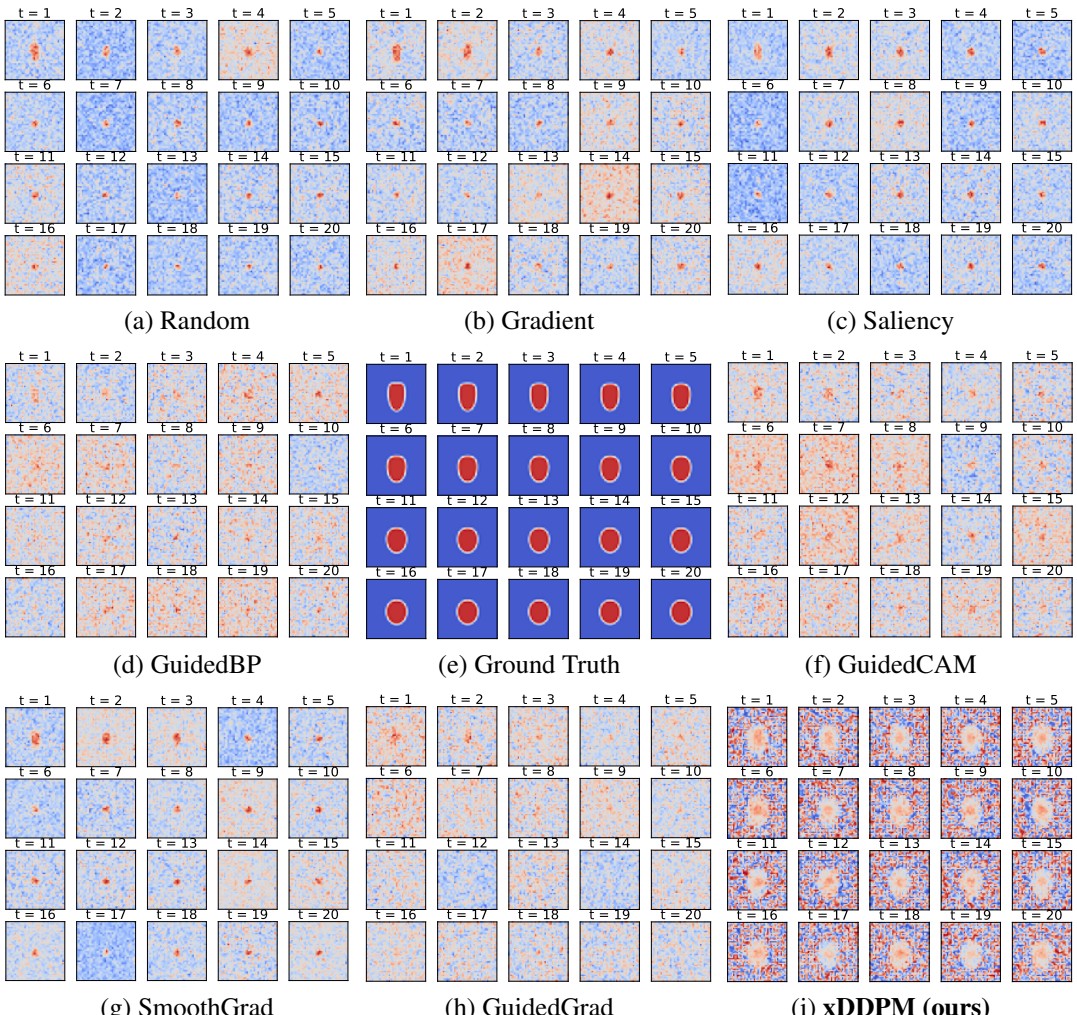

Figure 7: We present visualizations that compare the performance of our proposed method, xDDPM, with various baselines on the Wetting dataset. Specifically, we focus on the analysis of cells' circumference. The central portion of the visualizations represents the ground truth scenario, while the remaining figures depict the generated results from xDDPM and the baselines at different time steps. The use of mixed colors in the visualizations helps distinguish the irrelevant portions of the signal variable (cells' circumference), while the smoother and more uniform colors highlight the relevant parts. Methods that generate cell structures closely resembling the ground truth demonstrate higher explanatory capabilities. In this context, our xDDPM exhibits superior explanatory ability as the generated images at each time step closely resemble those of the ground truth.

on extensive biological and physical datasets, offering significant implications across diverse scientific and engineering fields. In the realm of biology, diffusion models aid in the modeling of protein structures and drug interactions. Additionally, in materials science, diffusion models contribute to the customization of material microstructures and properties. In the physical sciences, these models are employed to simulate the trajectories of moving objects. We believe that, endowed with the explanation generation capability of xDDPM, we can help gain more insight into the modeling of such systems.

Our approach combines the strengths of diffusion models and introduces the capability of generating samples with enhanced relevance to signals such as cell areas, cell circumferences, and forces of solid boundaries. However, there are currently certain limitations that need to be addressed. Firstly, our method xDDPM is currently exclusively applied to regression tasks and has not been adapted for classification tasks. It is an exciting future work to adapt it to the classification task, thus improving its generality. Additionally, there is room for improvement in terms of the scalability of our method, particularly when it comes to handling high-resolution samples (e.g., images or videos with $1024 \times 1024$ resolution). We intend to address these limitations in our future endeavors.

In conclusion, our approach makes the first step to imbue diffusion models with explanatory capabilities. Through the integration of the Information Bottleneck (IB) mechanism, our model, xDDPM, empowers the generation of samples that showcase heightened relevance to signal variances. We are confident that xDDPM will play a pivotal role in enhancing modeling techniques for a wide array of scientific applications.

