# OpenReview forum: "XDDPM: EXPLAINABLE DENOISING DIFFUSION PROB- ABILISTIC MODEL FOR SCIENTIFIC MODELING"
_ICLR.cc/2024/Workshop/AI4DiffEqtnsInSci — AI4DiffEqtnsInSci @ ICLR 2024 Poster_

### Official Review · Reviewer_gHQV · 2024-02-27
**Original and interesting approach to explainability in diffusion models**

**Rating:** 7
**Confidence:** 3

**Review:**

Summary:
The authors propose to incorporate an Information Bottleneck criterion into the training of diffusion models, to facilitate the generation of samples that contain an explanatory signal within the high-dimensional data space. They define the explanatory signal through a mask that focuses the denoising process only on those regions that are 'relevant'. The proposed IB approach is compared against several other attribution methods as a means to generate these masks.

Pros:
- Interesting and original framing of the question how to obtain explainable samples from diffusion models.
- Theoretically well-motivated objective, with both qualitatively and quantitatively convincing results.
- Scientifically relevant datasets.
- Motivation and theoretical background were presented in a way that was easy to follow.

Cons:
- The application of the baselines is left somewhat ambiguous, are they incorporated into the DDPM training instead of IB?
- The text would benefit from a more in-depth discussion of the experimental setup, perhaps in favor of a less theoretical perspective on IB?
- The utility of the method for scientific application is hard to judge from the provided experiments, could the authors present a case study of how they intend to utilize this method to address scientific questions?

---

### Official Review · Reviewer_PPM1 · 2024-02-27
**Relevant topic but diffuse and brief results and evaluation**

**Rating:** 4
**Confidence:** 3

**Review:**

### Summary:
An explainable DDPM is introduced to learn relevant components in three different tasks. Compared to other baselines, xDDPM is superior, yet several points remain unclear to me, as detailed below.

Overall, the paper organization and structure would benefit largely from a focus on nailing down the task concretely (what is the model supposed to produce) and describing how xDDPM accomplishes it.

### Strengths:
- Link to code and resources provided (not carefully checked, though)
- Nice to see how xDDPM finds the

### Weaknesses
- Given that "Random" is superior to many other methods, it appears unclear how suitable the other methods are in solving this task. Aren't there other methods in the literature that are better suited and allow for a stronger competition?
- The same argument also holds for the chosen metrics. For example, how does IoU measure whether the method extracts relevant components $S$ of the signal $X$? Also, it is unclear what goes into the metrics. Do you evaluate the generated mask $M$ or the actual output of your method?
- Unclear what price xDDPM introduces to the performance. That is, how accurate is xDDPM compared to DDPM in, e.g., fluid flow forecasting?
- Experiments is extremely vague/short and results section is missing.
- It remains unclear to me, how the method performs and what it is actually supposed to produce as output.

### Questions:
1. How does the forcing around the jelly fish-like robot look like in a normal DDPM?
2. How is mutual information, denoted as $I(\cdot;\cdot)$ defined?
3. What does $S$ represent intuitively? The pixels that change the most, or parts in the signal that effect other parts, or something completely different? Maybe you can provide some handy examples.
4. Wouldn't a baseline already perform well that computes parts in the scene that change least?

### Minor comments:
- Figures seem to lap over the left margin. Maybe correct for improved presentation.

---

### Meta-Review · Area_Chair_HAvQ · 2024-03-01

**Recommendation:** Accept (Poster)

**Metareview:**

This paper proposes to Explainable Denoising Diffusion Probabilistic Model (xDDPM) that enables the generation of samples in an explainable manner. The concerns raised by the reviewer PPM1 are valid, including missing experiments. However I notice that more information is included in the appendix. Although we have a page limit of 4, I strongly command the author to re-arrange the paper more properly and include more important content in the main text instead of leaving them in the appendix.

---

### Decision · Program_Chairs · 2024-03-01

Accept (Poster)